# Acute Kidney Injury: Medical Causes and Pathogenesis

**DOI:** 10.3390/jcm12010375

**Published:** 2023-01-03

**Authors:** Faruk Turgut, Alaa S. Awad, Emaad M. Abdel-Rahman

**Affiliations:** 1Faculty of Medicine, Mustafa Kemal University, Antakya 31100, Hatay, Turkey; 2Division of Nephrology, University of Florida, Jacksonville, FL 32209, USA; 3Division of Nephrology, University of Virginia, Charlottesville, VA 22908, USA

**Keywords:** acute kidney injury, prerenal AKI, acute tubular necrosis, acute interstitial nephritis, acute glomerulonephritis, postrenal AKI

## Abstract

Acute kidney injury (AKI) is a common clinical syndrome characterized by a sudden decline in or loss of kidney function. AKI is not only associated with substantial morbidity and mortality but also with increased risk of chronic kidney disease (CKD). AKI is classically defined and staged based on serum creatinine concentration and urine output rates. The etiology of AKI is conceptually classified into three general categories: prerenal, intrarenal, and postrenal. Although this classification may be useful for establishing a differential diagnosis, AKI has mostly multifactorial, and pathophysiologic features that can be divided into different categories. Acute tubular necrosis, caused by either ischemia or nephrotoxicity, is common in the setting of AKI. The timely and accurate identification of AKI and a better understanding of the pathophysiological mechanisms that cause kidney dysfunction are essential. In this review, we consider various medical causes of AKI and summarize the most recent updates in the pathogenesis of AKI.

## 1. Introduction

Acute kidney injury (AKI) is a frequent medical challenge associated with increased mortality, prolonged hospital stay, and the risk of chronic kidney disease (CKD) [1,2]. The incidence of AKI varies depending on the care setting and the criteria used to define it. AKI occurs in 10–15% of patients admitted to the hospital and its incidence has been reported to be more than 50% in the intensive care units [3,4,5].

AKI occurring outside of the hospital setting is called community-acquired (CA)-AKI, and it has been reported a common event in the population [3]. Wonnacott et al. compared the epidemiology, risk factors and outcomes between patients with hospital-acquired (HA) AKI (*n* = 334) and CA-AKI (*n* = 686). They showed the incidence of CA-AKI hospital admissions to be almost double the incidence of patients with HA-AKI (4.3% vs. 2.1%) with similar risk factors. They further showed that, while CA-AKI patients have more severe AKI, they have similar renal outcomes and better survival [4].

The incidence of AKI has been increasing over the years, and many factors (aging population, the rise in predisposing comorbidities, increased use of nephrotoxic agents, and invasive procedures) may contribute to the increased incidence of AKI [5].

AKI defined by an abrupt decrease in kidney function that includes an increase in serum creatinine (≥0.3 mg/dL within 48 h or ≥1.5 times baseline) or urine volume < 0.5 mL/kg/h for 6 h [6]. Kidney disease is usually a silent condition, and its current diagnosis is dependent on interpreting the changes in kidney function or decreased urine output. Changes in kidney function are usually assessed by measuring solutes that are normally excreted by the kidney (creatinine and cystatin c) and by urine output over time. Unfortunately, both changes in serum creatinine and urine output are neither sensitive nor specific to AKI [7]. Changes in urine output might be more sensitive but appear less specific.

### 1.1. Subclinical AKI

In some AKI cases, kidney damage may develop, but clinical manifestations and kidney dysfunction may not be present, and it is called subclinical AKI [8]. These patients are most likely to have acute tubular necrosis yet not fulfilling the creatinine-based consensus criteria for AKI [9]. While subclinical AKI may never develop into functional AKI and the biomarkers immediately disappear after the insult has stopped, it may occur for a short period, and then functional AKI develops. Subclinical AKI can occur following certain insults, such as nephrotoxicity, NSAID and contrast dyes [10].

Clinicians must differentiate between functional changes of AKI with rising serum creatinine/decreasing urine output and structural kidney damage that may happen prior to the functional changes. This period of time between structural damage, as identified by biomarkers, and the functional changes occurring when there is a drop of more than 50% of GFR offers an opportunity for earlier intervention that can help in kidney recovery.

Biomarkers such as Cystatin C (CysC), neutrophil gelatinase-associated lipocalin (NGAL), and kidney injury molecule-1 (KIM-1) may be helpful for detecting an injury to the kidney well before the increase in serum creatinine during subclinical AKI [8,11]. The increase in biomarkers as NGAL and KIM-1 is associated with an increased risk of subsequent renal replacement therapy and/or mortality. CysC is a cysteine protease inhibitor with a small molecular weight of 13 kDa [12]. It is freely filtered in the glomerulus, almost completely reabsorbed and catabolized by proximal tubular cells with the kidney being the sole organ clearing it [13]. Thus, CysC is more accurate than creatinine in determining GFR. NGAL: is a 25 kDa protein that belongs to the lipocalin superfamily [14]. Haase et al. discovered that high NGAL levels were related to poor results even in the lack of diagnostic SCr elevations in a pooled data analysis of 10 studies of patients hospitalized to ICU [11,15]. KIM-1 is a 38.7 kDa type I transmembrane glycoprotein that is hardly expressed in normal animal kidney tissue but may be increased following kidney insults, and thus can serve as a marker of early kidney injury and predict adverse clinical outcomes [15,16].

### 1.2. AKI

The etiology of AKI is conceptually classified into three general categories: prerenal, intrarenal, and postrenal (Table 1). Although this classification may be useful in establishing a differential diagnosis, AKI is mostly multifactorial, and pathophysiologic features are shared among the different categories [7]. Other factors contribute to making AKI more complex as, in most cases, numerous factors contribute not only to AKI initiation but also to its progression. AKI may develop more commonly after exposure to certain insults or in susceptible groups and many common pathophysiological factors play into the pathogenesis of AKI (Figure 1). The main focus of this review is to summarize the various medical causes of AKI. Furthermore, we summarize the most recent research updates in the pathogenesis of AKI.

Prerenal disease and acute tubular necrosis are two major causes of AKI in hospitalized patients. Depending on the study, 25–60% of AKI cases are attributed to prerenal causes [17,18]. Maintaining a normal glomerular filtration rate (GFR) is dependent on adequate renal perfusion. Renal hypoperfusion can be a part of a generalized decrease in tissue perfusion or selective renal ischemia and plays a critical role in the pathogenesis of prerenal AKI. ‘True’ intravascular volume depletion or ‘effective’ intravascular volume depletion results in compromised renal perfusion. A variety of conditions cause prerenal AKI; volume depletion, impaired cardiopulmonary functions, renovascular disease, and intrarenal hemodynamic changes (Table 1). The use of medications that alter renal blood flow and intrarenal hemodynamics are also associated with AKI. In particular, antihypertensive medications (e.g., diuretics, angiotensin converting enzyme inhibitors, and angiotensin receptor blockers) can reduce intravascular volume, renal blood flow (RBF), and/or GFR.

The normal response of the kidney to decreased renal perfusion is to maximally concentrate the urine and reabsorb sodium to maintain or to increase intravascular volume and normalize renal perfusion. Autoregulatory mechanisms often compensate for some degree of reduced renal perfusion, and initially, the glomerular and tubular function remains normal. When the hypoperfusion is sustained or the adaptive response is inadequate, organ damage can occur. However, reduced RBF eventually leads to ischemia and cell death.

## 2. Specific Causes for Pre-Renal AKI

### 2.1. Cardio-Renal Syndrome Type 1

An acute worsening of cardiac function (type 1 cardio-renal syndrome) can lead to a reduction in effective circulatory volume and an increase in central venous pressure. Both result in inadequate RBF, which activates the renin angiotensin aldosterone system and the sympathetic nervous system. On the other hand, elevated central venous pressure results in renal venous hypertension and increased renal resistance, which causes a further deterioration in renal perfusion [19].

### 2.2. Hepatorenal Syndrome

Hepatorenal syndrome (HRS) is one of many potential causes of kidney dysfunction occurring in patients with hepatic cirrhosis. Based on the rapidity of the decline in kidney function, two types of HRS were identified. Type 1 is a rapidly progressive condition that leads to AKI, while renal function deteriorates slowly over weeks to months in type 2 hepatorenal syndrome [20]. HRS remains a diagnosis of exclusion. The kidneys do not sustain any structural damage and any identifiable cause of kidney dysfunction is not apparent.

The pathophysiology of HRS is not fully understood, but the complex interaction between several different factors is implicated [21]. Arterial vasodilation in the splanchnic circulation appears to play a central role. In patients with hepatic cirrhosis, vasodilators, particularly nitrous oxide, increase due to increased production and decreased hepatic clearance. Splanchnic vasodilation is an important factor causing reduced effective circulating volume, resulting in renal hypoperfusion [21,22]. Cardiomyopathy related to hepatic cirrhosis may also contribute to the development of the HRS. Certain precipitating factors (spontaneous bacterial peritonitis, gastrointestinal bleeding, or infections) can further disturb the delicate compensatory hemodynamic balance and can be identified in some patients with hepatorenal syndrome.

## 3. Intrarenal Causes of AKI

Intrarenal causes of AKI can be categorized based on the primarily affected components of the kidney, including glomeruli, interstitium, tubules, or vascular components (Table 1).

### 3.1. Acute Tubular Necrosis

Acute tubular necrosis (ATN) is the most common cause of intrarenal AKI in hospitalized patients. Renal ischemia, sepsis, and nephrotoxins are major causes of ATN. With prolonged ischemia, ATN can occur, which is also known as ischemic ATN. All conditions associated with the prerenal disease can cause ATN, but ischemic ATN may also occur in the absence of overt hypotension in conditions when renal autoregulation is impaired. Kidney damage most commonly occurs in patients with severe hypotension, especially in those with sepsis. Other causes of ATN include endogenous compounds (hemoglobin in hemolysis or myoglobin in rhabdomyolysis) and exogenous compounds (drugs or radiocontrast media) that directly damage renal tubules via several different mechanisms. Endothelial and epithelial cell injuries are main factors that contribute to the pathogenesis of ATN. ATN occurs in multiple phases. In the initial phase, prolonged hypoxia following an ischemic event causes injury in the renal endothelial and tubular epithelial cells with a combination of both necrosis and apoptosis. As the injury worsens, the release of pro-inflammatory molecules induces an inflammatory cascade. Necrotic cellular debris may cause intratubular obstruction. During the maintenance phase, cellular repair, apoptosis, and proliferation occur to maintain cellular and tubular integrity. In the recovery phase, blood flow returns to the normal range, and the cells re-establish intracellular homeostasis and polarity.

### 3.2. Sepsis-Associated AKI

Sepsis is the most common trigger of severe AKI in critically ill patients [23,24]. Sepsis associated with ATN is often related to severe and sustained prerenal factors. The underlying mechanisms also include the release of cytokines, kidney inflammation, and tissue edema. Endotoxemia causes the activation of vasoactive hormones, induction of nitric oxide synthase, the release of cytokines, and activation of neutrophils [25,26].

More specifically, sepsis can alter the immune system and immune responses, with activation of the complement system and cellular innate immunity leading to the cell apoptosis/death pathway. The complement system can be directly or indirectly activated in patients with sepsis and contributes the pathogenesis. Sepsis was also found to alter microcirculation flow with kidney tissue hypoperfusion and hypoxia. Other mechanisms suggested to explain sepsis-induced AKI include the release of micro-RNAs causing cell death by repressing target protein expression at the post-transcriptional level and efferocytosis (clearance of dead cells by phagocytes) activation by sepsis leading to an increase in inflammatory mediators by dying cells during AKI, which attracts infiltrating immune cells and exacerbate tissue injury [27,28].

### 3.3. Rhabdomyolysis

Rhabdomyolysis is characterized by the breakdown of skeletal muscle resulting in the subsequent leakage of muscle cell contents (e.g., myoglobin, sarcoplasmic proteins, enzymes and electrolytes) into the extracellular fluid and circulation [29]. AKI due to rhabdomyolysis is quite common. Although the true incidence of AKI in rhabdomyolysis is difficult to establish, the reported incidence ranges from 13% to 50% of all cases. Myoglobin, creatine phosphokinase, and lactate dehydrogenase are the most important substances for muscle damage. These substances may be filtered through the glomeruli, leading to intratubular obstruction, inflammation, tubular damage, and renal vasoconstriction, and ultimately, the development of AKI [30]. Extracted fluid from the circulation into the swollen muscle groups leads to hypotension and shock.

### 3.4. Tumor Lysis Syndrome

Tumor lysis syndrome can occur before or after chemotherapy in patients with cancer. It is more common in rapidly growing hematologic malignancies, particularly during initial chemotherapeutic treatment. However, it can also be caused by solid organ tumors, although this is quite rare. The shift of electrolytes and nucleic acid material into the extracellular space may result in AKI. Acute uric acid nephropathy or parenchymal and tubular deposition of calcium phosphate crystals are the main factors causing AKI [31].

## 4. Contrast-Induced AKI

AKI may occur following the intravenous administration of contrast agents. Several mechanisms can contribute to the contrast agent nephrotoxicity. Suggested mechanisms include renal ischemia, vasoconstriction, formation of reactive oxygen species and direct tubular toxicity [32]. AKI that is secondary to contrast agents may present a mixed and picture of prerenal with the vasoconstriction ATN with the tubular toxicity.

### 4.1. Myeloma-Cast Nephropathy

Multiple myeloma is a cancer of plasma cells, with almost half of these patients having some sort of kidney-related disease and up to 20% have severe AKI. Severe AKI is usually a consequence of myeloma cast nephropathy, caused by high levels of immunoglobulin-free light chains.

Furthermore, several other factors can contribute to AKI in patients with multiple myeloma. The association between multiple myeloma with infection, dehydration, hypercalcemia and bone pain, leading to prescription of non-steroidal anti-inflammatory drugs can all contribute to AKI in these patients [33].

#### 4.1.1. Acute Interstitial Nephritis

Acute interstitial nephritis (AIN) can be secondary to many conditions. Medications are the most common cause, but acute interstitial nephritis may also be caused by infection, infiltrative disease, or simply idiopathic disease. Many drugs can cause acute interstitial nephritis, and the most common drugs are proton pump inhibitors, NSAID, penicillin, and cephalosporin (Table 2).

Drug-induced AIN is produced by idiosyncratic delayed type IV hypersensitivity reaction, but the precise pathophysiological mechanism is not fully understood.

#### 4.1.2. Renal Parenchymal Disease

Renal parenchymal diseases other than ATN is another important cause of intrinsic AKI. In general, the two types of renal parenchymal diseases that cause AKI are rapidly progressive glomerulonephritis and acute proliferative glomerulonephritis. Glomerulonephritis accounts for about 10% of AKI [34]. A nephritic or nephrotic pattern is observed in renal parenchymal diseases. Acute glomerulonephritis, characterized by proteinuria, hematuria, and hypertension (nephritic pattern) frequently causes AKI. A nephrotic pattern indicates non-proliferative glomerulopathy and is a rare cause of AKI. Most cases of glomerulonephritis are caused by an autoimmune response. The immunologic responses trigger an inflammatory process (e.g., complement activation, leukocyte migration and release of growth factors and cytokines) and the proliferation of glomerular tissue that can result in damage to the basement membrane, capillary endothelium, or mesangial area. Inappropriate complement activation contributes to the pathogenesis of AKI [35]. Complement system activation is not only a proximal trigger of downstream inflammatory events, but may also account for the systemic inflammatory events in renal parenchymal diseases. Acute glomerulonephritis can be caused by a primary renal disease or as part of systemic disease (Table 3).

#### 4.1.3. Vascular Disease

Acute events involving small- and large-sized blood vessels can cause AKI. Nevertheless, the bilateral involvement or involvement of a solitary functioning kidney is necessary for AKI development in diseases affecting larger vessels.

Acute renal artery occlusion because of thrombosis or aortic dissection results in a renal infarct. Emboli originating from the heart, or the aorta (athero-emboli) may also lead to renal artery obstruction. Thrombosis may spontaneously occur in a renal artery or after trauma, surgery or angiography. Acute renal vein thrombosis is a rare clinical entity and is generally associated with nephrotic syndrome.

Small vessel diseases include vasculitis and diseases, presenting with thrombotic micro-angiopathy. Small vessel vasculitis usually present as rapidly progressive glomerulonephritis [36]. Thrombotic micro-angiopathies occur in a range of conditions, including thrombotic thrombocytopenic purpura, hemolytic uremic syndrome, HELLP (hemolysis, elevated liver enzymes, and low platelets) syndrome, and malignant hypertension [37]. In many cases, these pathogenetic mechanisms are multifactorial in thrombotic micro-angiopathies. Thrombi in capillaries and arterioles and endothelial injury are characteristic findings of renal biopsy.

Renal atheroembolic disease is another common cause of vascular damage. During angiography or angioplasty, catheter manipulations disrupt the atheroma plaques, exposing the soft, cholesterol-laden core of the plaque to the arterial circulation [38]. In the same way, during surgical procedures, mechanical trauma (manipulation of the vessel or clamping) may also disrupt the atherosclerotic plaques. Irregularly shaped cholesterol crystal emboli can cause partial or complete obstruction of small renal arteries in atheroembolic renal disease. This causes distal ischemia and is followed by more tissue necrosis and inflammation. Furthermore, the renin angiotensin aldosterone system and complement activation also contribute to the development of AKI.

## 5. Specific Etiologies of AKI Affecting Multiple Renal Structures

### 5.1. Medications

AKI can be caused by various medications, but certain medications are more likely to cause AKI (Table 2). AKI associated with medications is reported in 14.4–37.5% of adults in different studies, depending on the definition and study population [24,39]. Medications induce various forms of kidney injury. The tubule–interstitial compartment is the most commonly affected area in the kidney. Acute interstitial nephritis is the most commonly reported histological manifestation of drug-induced AKI. Immune-mediated infiltration of immune cells in the tubule–interstitium or intratubular crystal deposition induced inflammation and renal hemodynamic deterioration are putative mechanisms in the pathogenesis [40].

AKI may develop after the administration of iodinated contrast media. However, it is not possible to exclude other causes of AKI in patients who developed AKI after contrast media administration. Therefore, AKI occurring after the administration of iodinated contrast media is currently called ‘contrast-associated AKI’ or ‘post-contrast AKI’. Studies reported evidence of ATN. Alterations in vasoactive peptides result in renal vasoconstriction and medullary hypoxia. The cytotoxic effects of the contrast agents on tubular cells also contribute to ATN [41,42].

### 5.2. Infections

Infections cause several forms of renal injuries, including AKI, acute or chronic glomerulonephritis, and tubule–interstitial nephritis. Bacterial, viral, and other infections may induce AKI. Bacterial pyelonephritis can cause AKI only if it is severe and bilateral. Kidney damage may occur through a direct invasion by the offending microorganisms or immune mechanisms involving microbial organisms. Kidney injuries may also occur as a part of sepsis-associated multi-organ failure [43].

### 5.3. COVID-19-Associated AKI

In patients with COVID, kidney involvement is frequent and associated with worse outcomes [44]. Clinical presentation ranges from mild proteinuria to progressive AKI. ATN and glomerular injury seem to be the most common cause of AKI [45,46]. It is largely unknown whether AKI is due to hemodynamic changes and cytokine release or direct viral cytotoxicity. The enhanced release of inflammatory mediators can play a key mechanism in tissue damage of patients with COVID-19. Acute tubular, glomerular and endothelial injuries are the most common histological findings. Thrombotic microangiopathy findings were also observed in kidneys of patients who died from COVID-19. Non-specific factors, such as mechanical ventilation, hypoxia, hypotension, low cardiac output and nephrotoxic agents may also contribute to kidney injury in severely affected patients [47]. Similar to previously reported descriptions in other viral infections, collapsing glomerulopathy has also been reported in patients with COVID-19 and has been described as COVID-19-associated nephropathy (COVAN) [47]. The carriers of apolipoprotein L1 (APOL1) risk variants are considered to be at particular risk of COVAN [47,48].

### 5.4. Post-Renal Causes of AKI

Anatomic obstruction of the urinary system at any level may result in AKI. Urinary tract obstruction is commonly caused by stones, external compression by tumors, retroperitoneal fibrosis, and bladder or prostate disorders (Table 1). Between 5 and 10% of all AKI episodes are caused by urinary tract obstruction [17]. This incidence increases in the elderly, but it is lower in patients with hospital-acquired AKI than in those with community-acquired AKI [17,23,49]. Benign prostatic hyperplasia is the most common cause of obstruction in older men. Impaired kidney function due to urinary tract obstruction is called obstructive nephropathy.

For AKI to develop, bilateral ureter obstruction or unilateral obstruction in a single functioning kidney or obstruction below the bladder must occur. Either complete or incomplete obstruction may cause AKI. A complete obstruction causes anuria, but incomplete obstruction may be associated with varying urine volumes from low to normal or polyuria. In patients with unilateral obstruction, serum creatinine levels usually remain normal.

Though the exact pathogenesis of AKI from urinary obstruction is not fully understood, various mechanisms have been identified in experimental studies. Following acute obstruction, the intraluminal pressure is directly reflected in the tubules and Bowman’s space within the first few hours. Subsequently, filtration pressure and capillary permeability decrease in the glomeruli. After 2–3 h of obstruction, renal plasma flow increases due to prostaglandins to maintain GFR by overcoming the elevated intratubular pressure. However, after 24–48 h, vasoconstriction in afferent and efferent arterioles occurs and renal plasma flow decreases due to the increase in thromboxane A2 [50,51]. For the detection of obstructions, the radiographic imaging of the urinary collecting system is essential [52,53]. The extent of renal recovery is influenced by the duration and severity of obstruction.

## 6. Conclusions

AKI often has more than one cause. Decreased renal perfusion is the most common cause of community acquired AKI and ATN is the most common cause of AKI in hospitalized patients. All clinical phenotypes of AKI cannot share a single pathophysiologic pathway. AKI is generally asymptomatic until the late stage. The past medical history, the timing of onset of AKI, and detailed physical examination are particularly important to identify the underlying etiology. Identifying causes is crucial because timely treatment can reverse the decline in GFR.

## Figures and Tables

**Figure 1 jcm-12-00375-f001:**
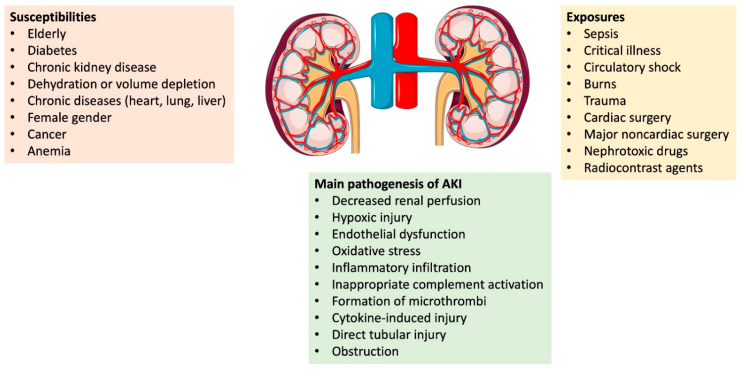
Risks and Pathogenesis of AKI.

**Table 1 jcm-12-00375-t001:** Medical causes of acute kidney injury.

Category	Abnormality	Possible Causes
**Prerenal**	True volume depletion	Hemorrhage
		Poor oral intake
		Gastrointestinal losses (vomiting, diarrhea)
		Third space losses (pancreatitis, peritonitis, burns)
		Renal losses (over diuresis)
		Skin or respiratory losses
	Impaired cardiopulmonary functions	Congestive heart failure
		Pericardial tamponade
		Pulmonary thromboembolism
	Decreased vascular resistance	Systemic vasodilation
		Sepsis
		Neurogenic shock
		Anaphylaxis
		Hepatorenal syndrome
	Intrarenal hemodynamic changes	Medications (NSAID, RAS blockers, CNIs)
		Hypercalcemia
**Intrinsic**	Tubular damage	Renal ischemia
		Nephrotoxins
		Endogenous
		Myoglobin, hemoglobin
		Tumor lysis syndrome
		Exogenous
		Medications (e.g., contrast agents)
	Glomerular damage	Acute glomerulonephritis
		Vasculitis
		Malign hypertension
		Thrombotic microangiopathies
	Interstitial damage	Infections (Bacterial or viral)
		Medications (Antibiotics, NSAIDs)
	Vascular damage	Renal artery/vein thrombosis
		Vasculitis (Polyarteritis nodosa)
		Atheroembolism
**Postrenal**	Intrarenal obstruction	Nephrolithiasis
	Extrarenal obstruction	Benign prostate hypertrophy
		Ureterolithiasis
		Prostate, bladder, rectal or cervical cancer
		Acute neurogenic bladder
		Urethral stenosis or clotting
		Retroperitoneal fibrosis
		Renal papillary necrosis

NSAID: Non-steroidal anti-inflammatory drugs, RAS: Renin angiotensin aldosterone system, CNIs: Calcineurin inhibitors.

**Table 2 jcm-12-00375-t002:** Medications associated with AKI.

**Antibiotics**	Aminoglycosides (Tobramycin, gentamycin)
	Vancomycin
	β-Lactam antibiotics
	Fluoroquinolones
	Rifampin
**Antiviral agents**	Tenofovir
	Cidofovir
	Foscarnet
	Acyclovir
	Indinavir
**Antifungals**	Amphotericin B
**Analgesics**	NSAIDs (Naproxen, ibuprofen)
**Chemotherapeutic agents**	Cisplatin
	Ifosfamide
	Tyrosine kinase inhibitors
	PD-1/PD-L1 inhibitors
	CTLA-4 inhibitors
**Other agents**	Lithium
	Phenytoin
	Proton pump inhibitors
	Furosemide
	Zoledronic acid
	Intravenous immunoglobulin
	Iodinated contrast media

NSAID: Non-steroid anti-inflammatory drugs, PD-1: Programmed cell death protein 1, PD-L1: Programmed death ligand 1, CTLA-4: T-lymphocyte-associated protein 4.

**Table 3 jcm-12-00375-t003:** Acute glomerular diseases causing AKI.

**Systemic disease**	Diffuse proliferative lupus nephritis
	ANCA-associated vasculitis
	Goodpasture’s syndrome
	Thrombotic microangiopathies (HUS/TTP)
	Polyarteritis nodosa
	Cryoglobulinemia
**Renal disease**	Anti-glomerular basement membrane disease
	Post-infectious glomerulonephritis
	Membranoproliferative glomerulonephritis
	IgA nephropathy

HUS: Hemolytic uremic syndrome, TTP: Thrombotic thrombocytopenic purpura. ANCA: Antineutrophil Cytoplasmic Antibodies.

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
