# Peer review of "Acute Kidney Injury: Medical Causes and Pathogenesis"

_jcm, 2023, doi:10.3390/jcm12010375_

Round 1

Reviewer 1 Report

I would like to summarize the notable findings of my review as follows:

Strengths of the article

1. The article presents a neat and well organized lay out on the topic and authors have not deviated from the main theme of the paper.

2. They have performed a good job in condensing a fairly big topic into its current format.

Drawbacks: 

1. Little information on the definition of AKI which should be within the scope of this review. 

2. The article provides a very basic information on the various etiologies and pathogenesis of AKI. This largely relates to the fact that this is a vast topic to cover and the information provided by authors is already well described in text books or other articles, hence, it is unlikely to provide additional insight into this topic. 

3. Given the limited scope of the topic, readers would not gain any insight into topics necessary for further management of AKI in clinical setting such as lab work up of various etiologies, diagnostic challenges and role of biomarkers. 

4. Covid-19 associated AKI and glomerulopathy is an evolving topic. The authors have provided limited information on its association with APOL-1 mutations 

Author Response

We thank you very much for allowing us to revise our manuscript, we appreciate the reviewers very much for their positive and constructive comments and suggestions on our manuscript entitled ‘Acute kidney injury; Medical causes and pathogenesis.

We have studied the reviewer’s comments carefully and have made revisions in the manuscript with track changes. We have tried our best to revise our manuscript according to the comments, hoping it will meet with approval. The responses are as follows,

Reviewer 1

Strengths of the article

  1. The article presents a neat and well organized lay out on the topic and authors have not deviated from the main theme of the paper.
  2. They have performed a good job in condensing a fairly big topic into its current format.

Response: Thank you for your encouraging comments.

Drawbacks:

  1. Little information on the definition of AKI which should be within the scope of this review.

Response: We have added the definition of AKI.

  1. The article provides a very basic information on the various etiologies and pathogenesis of AKI. This largely relates to the fact that this is a vast topic to cover and the information provided by authors is already well described in text books or other articles, hence, it is unlikely to provide additional insight into this topic.

Response: Agree with the reviewer.

  1. Given the limited scope of the topic, readers would not gain any insight into topics necessary for further management of AKI in clinical setting such as lab work up of various etiologies, diagnostic challenges and role of biomarkers.

Response: The present focus of the present manuscript is to shed light on the diverse medical causes of AKI and pathogenesis. Therefore, we did not provide any information regarding differential diagnosis and management of AKI.

  1. Covid-19 associated AKI and glomerulopathy is an evolving topic. The authors have provided limited information on its association with APOL-1 mutations

Response: We agree with the reviewer. We added some more data regarding COVID associated AKI and APOL-1 mutations.

Reviewer 2 Report

Dear Authors, nice and comprehensive manuscript, however very little new insights in AKI apart of Covid-19. You mention several times a complement involvement in the pathogenesis of AKI. I suggest more information on this aspect of AKI mechanisms could be beneficial for the paper. Particularly, in kidney transplantation contemporary complement blocking medications was effective in prevention/amelioration the reperfusion injury which is one of `the guise of AKI   

Author Response

We thank you very much for allowing us to revise our manuscript, we appreciate the reviewers very much for their positive and constructive comments and suggestions on our manuscript entitled ‘Acute kidney injury; Medical causes and pathogenesis.

We have studied the reviewer’s comments carefully and have made revisions in the manuscript with track changes. We have tried our best to revise our manuscript according to the comments, hoping it will meet with approval. The responses are as follows,

Reviewer 2

Dear Authors, nice and comprehensive manuscript, however very little new insights in AKI apart of Covid-19. You mention several times a complement involvement in the pathogenesis of AKI. I suggest more information on this aspect of AKI mechanisms could be beneficial for the paper. Particularly, in kidney transplantation contemporary complement blocking medications was effective in prevention/amelioration the reperfusion injury which is one of `the guise of AKI  

Response: Agree with reviewer. We added a few sentences regarding complement system activation in AKI pathogenesis.

Reviewer 3 Report

The manuscript summarized the most recent updates in the pathogenesis of AKI, mainly the medical causes. The content is quite comprehensive, but there are still some concerns that should be well addressed.

1. Is there any correlation between prerenal, intrarenal, and postrenal causes of AKI?

2. For each category, what is the specific molecular mechanism? This should be discussed and given a figure.

3. A summary figure is recommended.

Author Response

We thank you very much for allowing us to revise our manuscript, we appreciate the reviewers very much for their positive and constructive comments and suggestions on our manuscript entitled ‘Acute kidney injury; Medical causes and pathogenesis.

We have studied the reviewer’s comments carefully and have made revisions in the manuscript with track changes. We have tried our best to revise our manuscript according to the comments, hoping it will meet with approval. The responses are as follows,

Reviewer 3

The manuscript summarized the most recent updates in the pathogenesis of AKI, mainly the medical causes. The content is quite comprehensive, but there are still some concerns that should be well addressed.

  1. Is there any correlation between prerenal, intrarenal, and postrenal causes of AKI?

Response: Prerenal AKI is mainly secondary to renal hypoperfusion, which if prolonged can lead to tubular ischemia and acute tubular necrosis.

  1. For each category, what is the specific molecular mechanism? This should be discussed and given a figure.

Response: We have added a figure explaining the pathogenesis briefly.

  1. A summary figure is recommended.

Response: Causes of AKI were summarized in Table 1.

Round 2

Reviewer 1 Report

The script provides a general overview of the etiology and pathogenesis of AKI. The authors have made modifications the manuscript based on the some suggestions offered by the reviewers. The contents of the script lack novel information. 

Author Response

Thank you. There were no more critiques. Appreciate your prior comments/critiques.

Reviewer 3 Report

I have no further comments

Author Response

Repose to Critique

Thank you. There were no more critiques. Appreciate your prior comments/critiques.

Emaad Abdel-Rahman, MD
